# Role of Intestinal Dysbiosis and Nutrition in Rheumatoid Arthritis

**DOI:** 10.3390/cells11152436

**Published:** 2022-08-05

**Authors:** Malavikalakshmi Attur, Jose U Scher, Steven B. Abramson, Mukundan Attur

**Affiliations:** 1Drexel University College of Medicine, Drexel University, Philadelphia, PA 19129, USA; 2Division of Rheumatology, Department of Medicine, NYU Grossman School of Medicine, NYU Langone Health, New York, NY 10003, USA

**Keywords:** arthritis, microbiome, dysbiosis, nutrition

## Abstract

Rheumatoid arthritis is a chronic systemic immune-mediated disease caused by genetic and environmental factors. It is often characterized by the generation of autoantibodies that lead to synovial inflammation and eventual multi-joint destruction. A growing number of studies have shown significant differences in the gut microbiota composition of rheumatoid arthritis (RA) patients compared to healthy controls. Environmental factors, and changes in diet and nutrition are thought to play a role in developing this dysbiosis. This review aims to summarize the current knowledge of intestinal dysbiosis, the role of nutritional factors, and its implications in the pathogenesis of rheumatoid arthritis and autoimmunity. The future direction focuses on developing microbiome manipulation therapeutics for RA disease management.

## 1. Introduction

Rheumatoid arthritis (RA) is a chronic autoimmune disease found in roughly 1% of the population worldwide and that affects women more than men. It is characterized by polyarticular joint swelling, tenderness, and systemic inflammation due to the production of self-antigens (anti-rheumatoid factor or anti-citrullinated protein antibodies) in joint spaces. Substantial insights into RA pathophysiology suggest that various inflammatory pathways lead to an altered immune system and disease onset. The inflammatory pathway in RA is characterized by an imbalance in the anti-inflammatory—pro-inflammatory cytokine ratio, induced by alterations in the Th1 cell profile. There is release of IL-1, IL-6, and TNF-α from inflammatory cells in affected joint spaces, with concurrent decreases in IL-11, IL-13, and IL-10 [1]. Previously having high mortality and morbidity, the prognosis of RA has improved significantly in recent decades due to the development of more effective treatment options, including conventional disease-modifying anti-rheumatic drugs (DMARDs) and biologics that allow affected patients to achieve low disease activity or even clinical remission.

While the exact etiopathogenesis of RA is not well understood, it is currently thought that clinical presentation of RA results from genetically susceptible individuals coming into contact with environmental factors that induce targeted activation of the immune system, leading to disease. This theory is backed by monozygotic twin population studies showing low concordance rates between twins but high heritability rates [2,3,4]. Smoking is the environmental factor most consistently associated with RA development, but other factors include infections, diet, and perturbations in the oral and gut microbiota [5]. While significant research has focused on genetic factors, there is a growing realization that the host microbiota, especially the gut microbiota, play a crucial role in the development and progression of RA. In concert with the gut-associated lymphoid tissue, the gut microbiome is involved in maintaining immune homeostasis and acts as an indicator of the host’s health status. Perturbation of this interaction can affect mucosal and systemic immunity, and promote various inflammatory and autoimmune diseases [6,7,8].

Research in the microbiome field has exploded in recent decades. Now considered our “last organ” [9], it is clear that the micro-organisms in the human body are heavily tied to human health. Recent work has shown an increasing breadth of research connecting changes in the disturbances in the function of the microbiome, or dysbiosis, to the pathogenesis of RA. Consequently, attempts have been made to define the beneficial microbes and their metabolites for treating various diseases. Studies have also looked at nutritional modifications and their effect on the gut microbiota. This review will showcase current research regarding the nutrition–microbiome–disease axis and future research endeavors.

## 2. What Is the Gut Microbiome?

The microbiome is a term first coined by Whipps et al. in their work on rhizosphere micro-organisms, where they defined it as a “characteristic microbial community in a reasonably well-defined habitat that has distinct physiochemical properties as their theater of activity” [10]. Joshua Lederberg evolved this understanding and defined the microbiome as constituting the ecological communities of commensal, symbiotic, and pathogenic micro-organisms that share our body space. This interpretation allowed bacteria, fungi, protista, archaea, phages, viruses, and plasmids to all be members of a microbiome. Additionally, it is essential to distinguish between the terms microbiome and microbiota. While the microbiota is the collection of micro-organisms in a specified environment, the microbiome is the summation of all genetic material of the microbiota and their metabolic byproducts.

### 2.1. Development of the Gut Microbiome

The development of the microbiome starts at birth. The current consensus is that prior to birth, humans are sterile. While some studies contest this idea, noting the physiologic placenta and amniotic fluid colonization [11,12], more research must be done to validate the claim. In the immediate neonatal period, the infant’s intestine comes into contact with the maternal and external environment and is colonized. Depending on the mode of delivery, the microbiome’s composition will differ. Infants that are vaginally delivered are colonized with maternal vaginal *Lactobacillus*, *Prevotella,* and *Sneathia* on their skin and oral mucosa [13], and Enterobacteriaceae and breast milk Bifidobacteria colonize their intestine. Those delivered by C-section are colonized by more common skin microbes such as *Staphylococcus* [12]. Interestingly, a pilot study has shown that infants born via C-section and exposed to vaginal fluids at birth had microbiome compositions similar to those delivered vaginally, showcasing the importance of environmental exposure in the initial development of the microbiome.

The microbiome continues to change with time as the host ages. Development of the gastrointestinal system and the transition to an adult diet allows it to fully mature, becoming populated with the phyla *Firmicutes*, *Actinobacteria*, *Fusobacteria*, *Proteobacteria,* and *Bacteroidetes*, with the *Firmicutes* and *Bacteroidetes* representing over 90% of the population [14]. Table 1 further notes the common bacterial populations found in the gut microbiome. It is important to note that, even in healthy individuals, taxonomical variations exist due to differences in diet, geography, and lifetime medication and antibiotic use [15]. Despite these differences, the composition of the adult gut microbiome can be classified into three clusters, or enterotypes, based on species composition [14].

### 2.2. Gut Microbiome Function

The gut microbiota carries out a variety of physiological functions that help maintain health homeostasis, most of which revolve around the production of metabolites. With the consumption of food products, the microbial populations work to break down and derive energy from complex molecules, such as polysaccharides, lipids, and proteins. For example, Akkermansia is a bacterium commonly found in the human gut [17] that turns mucin into short-chain fatty acids (SCFAs), including acetate. Other beneficial bacteria such as Firmicutes use acetate to produce butyrate, a vital energy source for cells lining the gut. A recent study by Sun et al. reported a higher prevalence of the mucin-degrading Akkermansia in healthy subjects than in RA patients in the Chinese population [18]. Akkermansia is also inversely associated with obesity, diabetes, inflammation, and metabolic disorders. Response to fecal microbiota transplantation in treating irritable bowel syndrome also correlated with an increased relative abundance of Akkermansia. Reduction in the *Ruminococcus*, *Lachnospira*, and *Blautia genera*, as well as in the *Akkermansia muciniphila* species, is also reported in psoriasis [19,20,21], psoriatic arthritis, and spondyloarthritis [22], and negatively correlated with the serum levels of C3 in SLE patients [23].

A significant flow of carbon from dietary fibers to the host is through the transformation of non-digestible carbohydrates (e.g., cellulose and resistant starch), and through the fermentation process by the co-operative activity of commensal bacteria, into short chain fatty acids (SCFA) [24,25,26,27]. The major SCFA products are formate, acetate, propionate, and butyrate. Fermentation of protein-derived branched-chain amino acids also contributes to acetate and propionate production. Locally, SCFAs affect the intestine function via activation of Treg cells and can exert systemic effects following transport to other organs. A deficiency in SCFA production has been related to immune insufficiency and dysregulation in RA [28]. SCFA has a variety of functions. They activate signaling pathways via G-protein coupled receptors (GPCR), with three important ones being GPR41, GPR43, and GPR109A. Activation of GPR109A found on intestinal macrophages promotes anti-inflammatory properties, including the production of IL-10, and allows them to induce differentiation of T-regulatory cells [29]. The binding of acetate to GPR43 on intestinal epithelial cells (IEC) can cause hyperpolarization through potassium efflux, leading to the activation of the NLRP3 inflammasome complex [30]. The function of GPR41 was studied in GPR41/GPR43 KO mice, showing that exposure to oral infection, gut barrier disruption induced by ethanol, or TNBS-induced colitis led to increased expression of pro-inflammatory genes [31].

SCFA are also integral in aiding local cell proliferation. Germ-free (GF) or specific pathogen-free (SPF) mice showed a lower IEC proliferation ability than healthy controls. However, the proliferation rate improved when treated with SCFA and commensal bacterial populations. Conversely, SCFAs have been shown to suppress cancer cell proliferation and induce autophagy instead of apoptosis in colon cancer lines, to protect the local environment [32]. SCFA also functions in the stabilization of the gut epithelial barrier. Butyrate induces genes in the encoding of tight-junction proteins and stabilization of hypoxia-inducible factor-1 (HIF-1), which is important in maintaining barrier stability [33].

The gut microbiota also plays an essential role in nutritional homeostasis by producing essential vitamins, amino acids, and bile acids. For example, Bifidobacteria and Lactobacillus have been shown to be associated with increased intestinal barrier protection, and *Faecalibacterium prausnitzii*, *Roseburia*, *Bacteroides vulgatus*, and *Akkermansia muciniphila* are butyrate-producing bacteria. Vitamin K and the B group vitamins biotin, folate, niacin, pantothenate, pyridoxine, riboflavin, thiamine, and cobalamin are known to be produced by the microbiota. Vitamin K is essential for the normal function of the clotting cascade. GF mice grown without external vitamin supplementation were shown to have low PT levels and increased rates of hemorrhage events [34]. Many of the B vitamins are important in the stimulation of the immune system. Folate (B9) promotes the survival of T-regulatory cells through the downregulation of NK cell activity [35], while pyridoxine (B6) promotes the growth of *Bacteroides*, a key fighter of Salmonella typhimurium infections [36]. In addition, the production of vitamins such as menadione and coenzymes Q1-Q3 and 1,4-naphthoquinone helps inhibit pathogen growth and reduce exotoxin production by infecting Gram-positive bacteria [35]. While most of the amino acids in the intestines are derived from dietary sources, a small percentage are synthesized de novo by the gut microbiome; from 1 to 20% of circulating lysine originates from the gut microbiota [37]. Additionally, the production of D-amino acids is important in forming bacterial cell walls and inhibiting biofilm formation. D-serine can accumulate intracellularly during EHEC infections and suppress their type III secretion system, restricting colonization [38].

## 3. Microbiota in Rheumatoid Arthritis

Rheumatoid arthritis (RA) is a systemic autoimmune disease caused by genetic and environmental factors. Genetic research into rheumatoid arthritis is substantial; over 300 single-nucleotide polymorphisms (SNP) or risk loci have been found across all chromosomes in genome-wide association studies (GWAS), with roughly 100 being replicated in further meta-analyses [39]. However, the findings from these studies only explain a fraction of disease variance in patients and are not considered sufficient to guarantee disease development. The investigation was expanded and began to look at environmental factors, particularly the gut microbiome, as a critical factor in the etiology of RA. Hypotheses connecting the microbiome and RA are not novel. Andrews and Hoke associated “albuminous putrefaction” in the intestinal tract with the development of symptoms, and Hunter proposed the “oral sepsis hypothesis” that dental infections were correlated to RA [40]. Albuminous putrefaction (AP) is defined as the fermentation of water-soluble proteins present in food through the diversity of chemical reactions by bacteria in the intestine, thus leading to the formation of ptomaine ammonia compounds and hydrocarbons, which are known to alter the intestine integrity and gut microbiome.

Interestingly, Andrew and Hoke reported that the process of AP was present in rheumatoid patients and could be changed by correcting the type of food uptake. However, they were unclear whether the rheumatic condition occurred due to putrefaction or secondarily to the disease [41]. Regardless, a potential causal relationship is proposed in RA patients. With laboratory technological advances, research moved to animal models.

### 3.1. Experimental Animal Models

In one of the first studies done, Kohashi et al. found that in an adjuvant-induced arthritis model, GF rats developed severe arthritis, with a 100% incidence compared to SPF and conventional (CV) F344 rats. The study concluded that the presence of bacterial flora in the CV group modulated the development of the immune system, leading to a suppressed immune response with the induction of bacterial PG [42]. Other studies were done on the three main experimental animal models of arthritis: CIA, collagen-induced arthritis; K/BxN, mice expressing both the T cell receptor (TCR) transgene KRN and the MHC class II molecule A(g7); and SKG mice, which harbor a point mutation in the ZAP-70 gene, yielding reduced T-cell receptor (TCR) signaling.

### 3.2. CIA Model

Some studies utilized the CIA model, where injection of a type II collagen and Freund’s adjuvant emulsion leads to an inflammatory arthritic condition. In the initial stages of disease in affected mice, there are critical changes in the gut microbial populations. There is a decrease in *Bacteroidetes* and an increase in *Firmicutes* and *Proteobacteria*, all of which cause perturbations in metabolite production [43]. It is thought that the reduction in Bacteroidetes phylum leads to a reduction of CD4+ differentiation into T-_regulatory_ cells, resulting in an overall pro-inflammatory state. Interestingly, CIA mice grown in a GF environment that subsequently have *Bacteroides* introduced to the intestine showed an increase in T _regulatory_ (T-reg) cell population through stimulation of CD4+ cells by polysaccharide A, an immunomodulatory molecule released by *Bacteroides* [44]. A later study focused on the function of the intestinal barrier in this model. CIA mice exhibiting clinical disease had higher levels of zonulin, a protein essential in the modulation of tight junction permeability and previously shown to be more highly expressed in autoimmune conditions. Pre-treatment of mice with larazotide acetate, a zonulin antagonist, reduced arthritis onset [45].

### 3.3. SKG Model

In the SKG model, a point mutation in the Zap-70 gene leads to reduced T-cell signaling and altered thymic T-cell selection, causing spontaneous development of an inflammatory arthritic state [46]. When grown under GF conditions, the disease was not present. However, inoculation with *Prevotella* (discussed later) induced inflammatory arthritis [47].

### 3.4. K/BxN Model

Other studies utilized the K/BxN model, where mice expressed both the T-cell receptor (TCR) transgene KRN and MCH class II molecule A (g7). This led to the development of autoantibodies to glucose-6-phosphate and resultant severe inflammatory arthritis [48]. When grown in GF conditions, arthritis severity was significantly reduced, due to a decrease in autoantibody production and Th17 cells. The introduction of segmented filamentous bacteria (SFB) upregulates amyloid output in the ileum, inducing differentiation of Th17 cells and level reinstation, leading to arthritis [49]. SFB colonization generally induces a pro-inflammatory state. Small populations have been found in the human gut and are associated with increased IgA [50]. Flak et al. found increased gut permeability due to reduced numbers of tight junctions compared to healthy controls [51].

### 3.5. IL1rn -/- Model

Interleukin-1 receptor antagonist knock-out mice (il1rn -/-) spontaneously developed an autoimmune T-cell mediated arthritis, due to excessive IL-1 signaling. They did not develop the autoimmune disease when raised in a germ-free environment. Lack of IL-1RA reduces intestinal microbial diversity, specifically causing a decrease in *Ruminococcus* and *Prevotella* populations. This change causes an increase in Th17 cells, which are implicated in RA pathogenesis [52,53]. Interestingly, when the GF mice were colonized with *Lactobacillus bifidus*, they developed an arthritic state similar to the initially mentioned group, due to bacteria-induced activation of toll-like receptors (TLR) and a resultant imbalance in T-regulatory–Th17 homeostasis [54].

## 4. Human RA Studies

More recently, research has investigated the role of the gut microbiome flora and function in human RA. The 16S next-generation sequencing (NGS) allows for rapid comparison of gut flora from healthy controls to RA patients. *Prevotella copri* (*P. copri*) is patients’ main bacterial population associated with dysbiosis [55,56]. A first-degree human gut microbiome study from Rodriguez et al. showed that patients with preclinical auto-antibody positive RA had marked differences in gut microbial populations compared to healthy controls, namely an increase in *Prevotellacaea*, specifically *Prevotella* spp. [57]. Scher et al. found that *P. copri* was most abundant in new-onset RA compared to chronic RA, PsA, and healthy controls, and correlated with decreased *Bacteroides fragilis* [58]. Functional genomic analyses of these patients suggested a decrease in vitamin production due to the inability of *Prevotella* spp. to engage in vitamin metabolism. Intriguingly, the decrease in gut folate metabolism suggests an MTX treatment-resistant phenotype. Further studies may shed light on the drug metabolism capabilities of the gut microbiome, potentially affecting decision-making in disease treatment [59].

In addition, a later study found that *Prevotella copri* produced the Pc-p27 protein, which could trigger a Th1-mediated immune response by binding to HLA-DR in RA patients. This claim was supported by the IgA-like antibody response in acute and chronic RA patients to Pc-p27 and correlated with the production of TH17 cytokines and anti-citrullinated protein antibodies (ACPA) [60]. Another cross-sectional study from Chen et al. found lower gut microbial diversity in RA patients compared to healthy controls. Notably, RA patients on DMARDs such as methotrexate (MTX) and hydroxychloroquine (HCQ) had higher diversity than treatment-free individuals [8]. Other studies expanded on these findings, noting a decrease in *Faecalibacterium* and increases in *Collinsella* [61].

## 5. Rheumatoid Arthritis and Nutrition

As previously mentioned, environmental factors are thought to play a substantial role in the development of RA, with nutrition being one. Looking at worldwide dietary patterns, the heterogeneity in food choices, and an overall decrease in plant-based foods, with increased meat consumption and food processing [62], these dietary changes have resulted in observable changes in the gut microbiome function. One study comparing the microbiota of hunter-gatherers in Tanzania to healthy controls noted significantly higher levels of CAZymes, responsible for the breakdown and modification of oligo and polysaccharides [63]. Changes in the produced metabolites and enzymes showed that diet differences affect gut microbiome function. The conventional treatment of RA depends on using steroids, DMARD, and biological regimens along with appropriate healthy changes such as smoking cessation and weight reduction. As the field of research building on the relationship between the gut microbiome, diet, and autoimmune diseases increased, questions arose regarding the efficacy of dietary changes aimed at restoring health homeostasis. The Mediterranean diet has been the main target of study.

### 5.1. Mediterranean Diet

The Mediterranean diet, which utilizes olive oil as the principal source of fat in cooking, is characterized by a high consumption of vegetables, legumes, alcohol, and white meat (fish and chicken) and low salt. The diet excludes red meats, which are often processed with carcinogenic nitrites and pro-inflammatory nitrates. Studies have shown that long-term adherence to the diet leads to observable changes in the gut microbiome.

The Mediterranean diet (MED) has been associated with an over-representation of species such as *Faecalibacterium prausnitzii*, *Eubacterium*, and *Roseburia* [64], as well as *Collinsella aerofaciens*, *Dorea formicigenerans*, and *Clostridium ramosum*. MED also increased the abundance of *Prevotella* and branched-chain amino acids (BCAA) degrading pathways and decreased *Bifidobacterium* and BCAA biosynthesis pathways in the gut microbiome [65,66]. Other studies have reported an increased abundance of *Proteobacteria*, *Bacteroides uniformis*, and *Prevotella stercorea.* In contrast, western diets (characterized by high meat and fat intake) were associated with a decrease in gut microbiota diversity and an increase in the Firmicutes/Bacteroidetes ratio, which are related to many inflammatory-adjacent disorders such as type 2 diabetes and obesity [67]. Meslier et al. found that adopting the Mediterranean diet in obese and overweight patients for two months reduced cholesterol, increased insulin sensitivity, and enhanced gut microbial populations [68]. Other studies saw an upregulation in butyrate production, integral in immunological homeostasis [69,70].

As a result, the Mediterranean diet is frequently clinically recommended to patients due to its antioxidant [71,72] and immune-modulatory properties, stemming from the production of monounsaturated and polyunsaturated fatty acids (MUFA, PUFA) and polyphenols. Polyphenols’ production improves the intestinal barrier’s strengths through the upregulation of tight junction protein production [73]. Linoleic acid (LA) and alfa-linolenic acid (ALA) are essential fatty acids derived from the diet. Derived from n-6 and n-3 PUFA precursors, the ratios in which these compounds are found dictate their profound effects on immune modulation [74,75]. LA and n-6 PUFA come from animal products and are pro-inflammatory, while ALA and n-3 PUFA are found in fish, oil, nuts, and berries, and are anti-inflammatory [76]. Due to the composition of the Mediterranean diet, the n-6/n-3 ratio is balanced, unlike more western diets (heavy in fats, red meats, and food processing), where there is an excess of n-6 PUFA and a deficiency of n-3 PUFA metabolites [77]. Several case reports and cohort studies have suggested that the higher levels of n-3 PUFA have a protective modulatory role in RA-susceptible patients. However, further research is necessary [78,79].

### 5.2. Dietary Regimens

Other diets have been studied for the potential management of RA. Fasting has been shown to improve RA disease activity scores. Small cohort studies were initially performed and found that short periods of fasting led to improvements in disease activity scores [74,75]. This was further validated by Kragh et al., who placed patients on a 7–10 day-long fast that was transitioned into a lactovegetarian diet and found improvements in the number of tender and swollen joints, morning stiffness, pain scores, as well as ESR and CRP, with these changes lasting for up to a year post-study completion [76]. The elemental diet, a liquid meal replacement diet that holds its nutritional value as “dietary building blocks” (amino acids, vitamins, monosaccharides), was compared in a small study to 2-week oral prednisolone in RA patients and was found to be similarly effective in disease management [78]. Short-term high-fiber diet supplementation increased SCFA levels and decreased the synthesis of pro-inflammatory mediators, MCP-1, IL-18, and IL-33 [79].

### 5.3. Other Dietary Factors

Several studies have looked at more specific dietary choices and their effects. CIA mice models given small amounts of alcohol had the onset of arthritis delayed [80]. In a small cohort study, low to moderate consumption of alcohol in affected patients was found to be protective [81]. Regular consumption of sugary beverages was correlated with an increased risk of RA development, independent of obesity. High decaf coffee consumption, defined as >4 cups per day, was associated with increased RA onset, while tea consumption was inversely associated [82]. High salt consumption has been a risk factor, particularly in smokers. A study from Sweden showed that high sodium intake doubled RA risk in smokers [83], and a separate cohort study showed high daily sodium intake (>4.5 g per day) increased RA risk with adjustment for confounders such as HTN, CV disease, diabetes, and cancer [84,85].

## 6. Therapeutics Targeted toward Dysbiosis

As the microbiome–nutrition–disease axis becomes more evident in RA, attention has turned toward the potential of therapeutics for clinical management. Numerous emerging technologies aim to manipulate the gut microbiome, to induce a healthy state. Fecal material therapy (FMT) is one such technique. It involves the transfer of fecal material from a healthy donor to the gastrointestinal tract of a patient with dysbiosis. The fecal material can be administered via capsule, enema, colonoscopy, or nasoduodenal tube [86]. While FMT is best known for its treatment of chronic, recurrent *Clostridioides difficle* infection (*C. diff*), research is being conducted into its usage for other autoimmune disorders. Few studies exist looking at FMT usage in RA patients. A case report from Zeng et al. showed successful treatment of RA with FMT via a decrease in RF, DAS-28, and HAQ scoring immediately after a treatment lasting 78 days.

### 6.1. Probiotics

Probiotics, or live organisms intended to have health benefits with consumption, are also being researched [87]. Simply described, their intended function is to reduce harmful bacterial populations and increase beneficial ones, while maintaining an appropriate equilibrium. The most well-known probiotics include *Lactobacillus* and *Bifidobacterium*, populations that produce SCFA and that work to maintain gut mucosal stability [88]. Several studies have reported on the beneficial effects they had on animal models of arthritis. One study found that oral administration of *L. casei* in CIA mice delayed the onset and reduced the severity of CIA [89]. Fan et al. looked at the preventative vs. therapeutic administration of *B. adolescentis* in CIA mice [90]. Early administration showed a better promotion of SCFA production and reduction of clinical symptoms. Another study found that early administration of *L. helveticus* SBT2171 significantly reduced joint tenderness and swelling. These findings suggest prophylactic treatment may be more effective in disease management [91].

Unlike animal studies, research on the effect of probiotics on human disease states is limited. Hatakka et al. looked at the *L. rhamnosus* supplementation in stable RA patients not on DMARD mediations compared to healthy controls. Patients were given the intervention for 12 months, twice daily, in addition to their regular mediation regimen. While there were no significant differences in inflammatory markers or clinical disease compared to the placebo group, patients in the supplementation group noted a subjective health improvement [92]. Ramini et al. investigated the effect of probiotic supplementation on RA patients. Participants in the active arm were given an eight-week course of probiotics containing *L. acidophilus*, *L. casei*, and *B. bifidum*. Compared to the control group, they showed improvements in DAS28 and high sensitivity CRP (hs-CRP) levels. Other studies found similar improvements in clinical disease markers with *Lactobacillus* probiotic supplementation [93,94]. A recent study tested the effect of a combined probiotic pill containing *L. casei* LC-11, *L. acidophilus* LA-14, *Lactococcus lactis* LL-23, *B. lactis* BL-04, and *B. bifidum* BB-06 in active RA patients, compared to healthy controls. The test group showed significant reductions in WBC, TNF, and IL-6. However, no changes were noted in IL-10, adiponectin, CRP, and ESR levels [95].

### 6.2. Fermented Food

Fermented foods consist of foods or beverages made using the conversion of food components through microbial enzymes [96]. Dedicated research in the field is currently limited. However, observational studies have suggested that fermented foods positively modulate the gut immune system by forming bioactive metabolites in the present microbial populations [97]. *Bifidobacterium* and *Lactobacillus* are common microbial populations used in food preparation [98].

### 6.3. Microbiome-Derived Metabolites

The gut microbiome produces many bioactive metabolites, including vitamins, butyrate, and bile acids. As described earlier in the paper, these molecules work to maintain healthy homeostasis. Little research has been done to assess their potential use in microbiome modulation. Pena-Rodriguez et al. studied the effect of butyrate on the intestinal inflammatory response in a cholestasis model. Administration of butyrate was found to reduce pro-inflammatory cytokines and upregulate the production of tight junction proteins [99]. Berberine, a therapeutic option in treating type-2 diabetes, increased SCFA-producing bacteria and lowered gut inflammation [100].

## 7. Conclusions

Recent years have shown an explosion in the research supporting the gut microbiome’s role in RA pathogenesis. In a nutshell, the host–microbiota interactions and dysbiosis in rheumatoid arthritis are shown in Figure 1.

Experimental animal and human models indicate that dysbiosis occurs far before the clinical onset of disease and continues to shift during disease progression. However, to gain a mechanistic understanding of how the gut microbiota affects the host metabolism, comprehensive analyses of high-throughput multi-omics data, including metagenomics and metabolomics data, and measures of host physiology hold potential in the identification of possible molecular mechanisms in the host–microbiome association. Furthermore, we also increasingly understand how diet, through microbes or metabolites, modifies health. Clinical evidence needs to be translated into practice, using consistent matrices of prebiotics or probiotics or fecal microbiota transplantation to assess changes in gut microbiota composition and health outcomes. Connecting this knowledge to the importance of nutrition and diet in the modulation of the microbiome and the immune system has opened avenues of research into the therapeutic usage of interventions such as probiotics and microbial-derived metabolites in disease management. However, it is clear that further studies are needed before interventions can be considered for clinical use. One challenge is that many microbiome studies are more association studies than correlational ones.

Furthermore, the majority of studies are derived from sample repositories where the US is overrepresented [101]. As discussed, geography plays a prominent role in diet and lifestyle, affecting microbiome populations. More extensive and iterative clinical trials are necessary for study findings to have global generalizability. Another challenge involves the uncertainty regarding the future patentability of interventions. The patenting of natural products is controversial. A recent ruling in 2013 of the Association for Molecular Pathology vs. Myriad Genetics declared genes unpatentable due to their being products of nature [102]. How that affects bacterial strains’ patentability, or microbial metabolites remains unclear. Additionally, studies evaluating the host–microbiome relationship in RA are necessary to elucidate possible triggering factors and close the gap in our knowledge regarding the role of gene–environment interactions.

## Figures and Tables

**Figure 1 cells-11-02436-f001:**
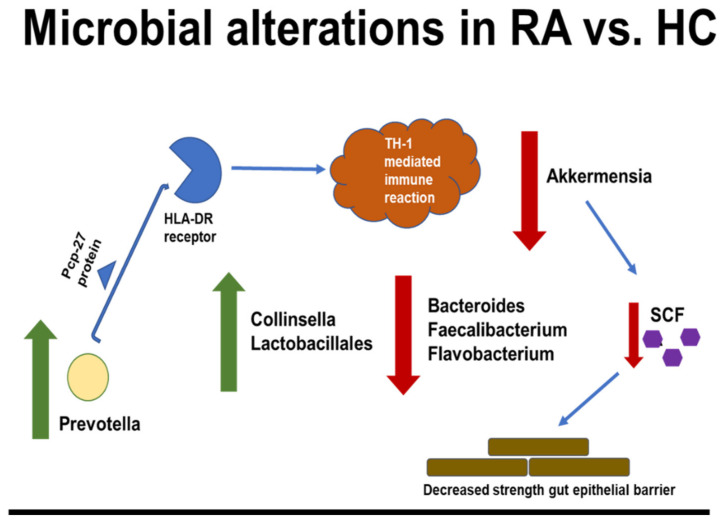
Microbial dysbiosis and gut short-chain fatty acid (SCFA) dysregulation lead to immune activation and inflammation in RA.

**Table 1 cells-11-02436-t001:** Taxonomic data of most common gut microbial populations [16].

Phylum	Class	Order
Firmicutes	Clostridia	Clostridiales
	Bacilli	Lactobacillales
	Erysipheoltrichia	Erisophelotrichales
	Negativicutes	Selenomonadales
		Veillonellales
Bacteroidetes	Bacteroidia	Bacteroidales
Actinobacteria	Coriobacteria	Coriobacteriales
		Eggerthellales
	Actinobacteria	Bifidobacteriales
Fusobacteria	Fusobacteria	Fusobacteriales
Proteobacteria	Gammaproteobacteria	Enterobacteriales
		Aeromonadales

## Data Availability

Not applicable.

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
