# Peer review of "Role of Intestinal Dysbiosis and Nutrition in Rheumatoid Arthritis"

_cells, 2022, doi:10.3390/cells11152436_

Round 1

Reviewer 1 Report

This is a nicely written narrative review on the role of intestinal dysbiosis and nutrition in RA. The review summarizes the most important findings to date and is up-to-date. I would suggest to add a picture or a table summarizing, for example, the current knowledge of microbiota alterations in RA vs HCs and/or another summarizing the studies that have been conducted so far.

Author Response

This is a nicely written narrative review on the role of intestinal dysbiosis and nutrition in RA. The review summarizes the most important findings to date and is up-to-date. I would suggest adding a picture or a table summarizing, for example, the current knowledge of microbiota alterations in RA vs HCs and/or another summarizing the studies that have been conducted so far.

Response: We appreciate the positive comments by the reviewer.  As recommended, we have added figure 1 summarizing the microbial alteration in RA.

Reviewer 2 Report

The paper is very interesting, actual and it opens new perspectives for the future. I suggest to the authors to tell about the role of Akkermansia in this disease and explain better, at page 3 in the chapter "Gut Microbiome Function" the degradation of complex carbohydrates in short - chain fatty acids.

At page 3, I suggest to the authors to explain better the transformation of complex carbohydrates in short-chain fatty acids

Always to page 3, at "the gut microbiota also plays......." I suggest to specify the bacteria that have a role in nutritional homeostasis.

At page 4, I would like to know more about "albuminous putrefaction" and explain better in the work.

At page 6, chapter 4.1 about Mediterranean Diet, I would like to know the bacteria involved

Author Response

Reviewer 2

The paper is very interesting, actual and it opens new perspectives for the future. I suggest to the authors to tell about the role of Akkermansia in this disease and explain better, at page 3 in the chapter "Gut Microbiome Function," the degradation of complex carbohydrates in short-chain fatty acids.

Response: Again, we would like to thank the reviewer for the comments and the opportunity to add to our work. Akkermansia is a mucin-degrading bacterium commonly found in the human gut [1] which turns mucin into short-chain fatty acids (SCFAs), including acetate. Other beneficial bacteria like Firmicutes use acetate to produce butyrate, a vital energy source for cells lining the gut. The bacterium is more abundant in the gut of healthy subjects than in diabetic and obese patients. A recent study by Sun et al. reported a higher prevalence of Akkermansia in healthy subjects than in RA patients in the Chinese population. Akkermansia is also inversely associated with obesity, diabetes, inflammation, and metabolic disorders. Response to fecal microbiota transplantation in treating irritable bowel syndrome also correlated with the increased relative abundance of Akkermansia [2-4]. Reduction in the Ruminococcus, Lachnospira, and Blautia genera, as well as in the Akkermansia muciniphila species, is also reported in psoriasis [2, 3], psoriatic arthritis, and spondyloarthritis [5] and negatively correlated with the serum levels of C3 in SLE patients [6]. Whereas, Akkermansia muciniphila was abundant (>2%) in enthesitis-related arthritis in children [7]. Stoll et al. have shown that Akkermansia may promote ankle arthritis in KRN/B6xNOD mice [8].

On page 3, I suggest to the authors to explain better the transformation of complex carbohydrates in short-chain fatty acids

Response: A significant flow of carbon from the dietary fibers to the host is through the transformation of non-digestible carbohydrates (e.g., cellulose and resistant starch) and through the fermentation process by the co-operative activity of commensal bacteria into short chain fatty acids (SCFA) [9-12]. The major SCFA products are formate, acetate, propionate, and butyrate. Fermentation of protein-derived branched-chain amino acids may also contribute to acetate and propionate production. Locally, SCFAs affect the intestine function via activation of Treg cells and can exert systemic effects following transport to other organs. SCFA deficiency has been related to immune insufficiency and dysregulation in RA [13].

Always to page 3, at "the gut microbiota also plays......." I suggest specifying the bacteria that have a role in nutritional homeostasis.

Response: Increased intestinal barrier protectors Bifidobacteria and Lactobacillus, butyrate-producing bacteria Faecalibacterium prausnitzii and Roseburia and Bacteroides vulgatus and Akkermansia muciniphila.

On page 4, I would like to know more about "albuminous putrefaction" and explain better in the work.

Response: Albuminous putrefaction (AP) is defined as the fermentation of water-soluble proteins present in food through the diversity of chemical reactions by bacteria in the intestine, thus leading to the formation of ptomaine, ammonia compounds, and hydrocarbons which are known to alter the intestine integrity and gut microbiome. Interestingly, Andrew and Hoke reported that the process of AP was present in rheumatoid patients and could be changed by correcting the type of food uptake. However, they were unclear as to whether the rheumatic condition occurred due to putrefaction or secondarily to the disease [14]. Regardless, a potential causal relationship is proposed in RA patients.

On page 6, chapter 4.1 about Mediterranean Diet (MED), I would like to know the bacteria involved

Response: The Mediterranean Diet (MED) has been associated with the over-representation of species like Faecalibacterium prausnitzii, Eubacterium, and Roseburia [15], Collinsella aerofaciens, Coprococcus comes, Dorea formicigenerans, Clostridium ramosum. MED diet is also reported to increase the abundance of Prevotella, and branched-chain amino acids (BCAA) degrading pathways and decrease Bifidobacterium and BCAA biosynthesis pathways in the gut microbiome [16], [17]. Other studies have reported an increased abundance of Proteobacteria, Bacteroides uniformis, and Prevotella stercorea. Families belonging to Firmicutes phyla, such as Ruminococcus and Lachnospiraceae, changed as well, with decreases in Roseburia, Roseburia faecis, and Pseudobutyrivibrio xylanivorans noted. In contrast, Western diets (characterized by high meat and fat intake) were associated with a decrease in gut microbiota diversity and an increase in the   Firmicutes/Bacteroidetes ratio, which are associated with many inflammatory disorders such as type 2 diabetes and obesity [18].

References:

  1. Zhou, K., Strategies to promote abundance of Akkermansia muciniphila, an emerging probiotics in the gut, evidence from dietary intervention studies. J Funct Foods, 2017. 33: p. 194-201.
  2. Schade, L., et al., The gut microbiota profile in psoriasis: a Brazilian case-control study. Lett Appl Microbiol, 2022. 74(4): p. 498-504.
  3. Scher, J.U., et al., Decreased bacterial diversity characterizes the altered gut microbiota in patients with psoriatic arthritis, resembling dysbiosis in inflammatory bowel disease. Arthritis Rheumatol, 2015. 67(1): p. 128-39.
  4. Hamazaki, M., et al., Fecal microbiota transplantation in the treatment of irritable bowel syndrome: a single-center prospective study in Japan. BMC Gastroenterol, 2022. 22(1): p. 342.
  5. Wang, L., et al., Gut microbiota changes in patients with spondyloarthritis: A systematic review. Semin Arthritis Rheum, 2022. 52: p. 151925.
  6. Liu, F., et al., Distinct Microbiomes of Gut and Saliva in Patients With Systemic Lupus Erythematous and Clinical Associations. Frontiers in Immunology, 2021. 12.
  7. Stoll, M.L., et al., Altered microbiota associated with abnormal humoral immune responses to commensal organisms in enthesitis-related arthritis. Arthritis Res Ther, 2014. 16(6): p. 486.
  8. Stoll, M.L., et al., Akkermansia muciniphila is permissive to arthritis in the K/BxN mouse model of arthritis. Genes and immunity, 2019. 20(2): p. 158-166.
  9. den Besten, G., et al., The role of short-chain fatty acids in the interplay between diet, gut microbiota, and host energy metabolism. J Lipid Res, 2013. 54(9): p. 2325-40.
  10. Morrison, D.J. and T. Preston, Formation of short chain fatty acids by the gut microbiota and their impact on human metabolism. Gut Microbes, 2016. 7(3): p. 189-200.
  11. Ohland, C.L. and C. Jobin, Microbial Activities and Intestinal Homeostasis: A Delicate Balance Between Health and Disease. Cellular and Molecular Gastroenterology and Hepatology, 2015. 1(1): p. 28-40.
  12. Belkaid, Y. and O.J. Harrison, Homeostatic Immunity and the Microbiota. Immunity, 2017. 46(4): p. 562-576.
  13. Park, J., et al., Short-chain fatty acids induce both effector and regulatory T cells by suppression of histone deacetylases and regulation of the mTOR–S6K pathway. Mucosal Immunology, 2015. 8(1): p. 80-93.
  14. Andrews, C.R.a.H., M, A preliminary report on the relation of albuminous putrefaction in the intestines to arthritis deformans (rheumatoid arthritis, osteo-arthritis); its influence upon treatment. . Am. J. Orthop. Surg. , 1907-08: p. 61-72.
  15. Ghosh, T.S., et al., Mediterranean diet intervention alters the gut microbiome in older people reducing frailty and improving health status: the NU-AGE 1-year dietary intervention across five European countries. Gut, 2020. 69(7): p. 1218-1228.
  16. Rinott, E., et al., The effects of the Green-Mediterranean diet on cardiometabolic health are linked to gut microbiome modifications: a randomized controlled trial. Genome Medicine, 2022. 14(1): p. 29.
  17. Nagpal, R., et al., Gut microbiome-Mediterranean diet interactions in improving host health. F1000Res, 2019. 8: p. 699.
  18. Tsigalou, C., et al., Gut microbiome and Mediterranean diet in the context of obesity. Current knowledge, perspectives and potential therapeutic targets. Metabolism Open, 2021. 9: p. 100081.

Reviewer 3 Report

Dear Authors,

Please improve and extent your introduction

Please add more conclusion to your review

Kind regards

Author Response

Reviewer 3

Please improve and extent your introduction and add more conclusions to your review.

Response: As recommended by the reviewer, we have expanded the introduction and conclusion sections.